# The spatio-temporal features of perceived-as-genuine and deliberate expressions

**Shushi Namba**[1]*, **Koyo Nakamura**[2,3,4], **Katsumi Watanabe**[4]

**1** Psychological Process Research Team, Guardian Robot Project, RIKEN, Kyoto, Japan, **2** Faculty of Psychology, Department of Cognition, Emotion, and Methods in Psychology, University of Vienna, Vienna, Austria, **3** Japan Society for the Promotion of Science, Tokyo, Japan, **4** Faculty of Science and Engineering, Waseda University, Tokyo, Japan

* sushishushi760@gmail.com

**Data Availability Statement:** Data can be found at OSF: (https://osf.io/e7pdt).

**Funding:** This research was supported by Early-Career Scientists (20K14256) from JSPS to S. N., Early-Career Scientists (19K20387) from JSPS to

## Abstract

Reading the genuineness of facial expressions is important for increasing the credibility of information conveyed by faces. However, it remains unclear which spatio-temporal characteristics of facial movements serve as critical cues to the perceived genuineness of facial expressions. This study focused on observable spatio-temporal differences between perceived-as-genuine and deliberate expressions of happiness and anger expressions. In this experiment, 89 Japanese participants were asked to judge the perceived genuineness of faces in videos showing happiness or anger expressions. To identify diagnostic facial cues to the perceived genuineness of the facial expressions, we analyzed a total of 128 face videos using an automated facial action detection system; thereby, moment-to-moment activations in facial action units were annotated, and nonnegative matrix factorization extracted sparse and meaningful components from all action units data. The results showed that genuineness judgments reduced when more spatial patterns were observed in facial expressions. As for the temporal features, the perceived-as-deliberate expressions of happiness generally had faster onsets to the peak than the perceived-as-genuine expressions of happiness. Moreover, opening the mouth negatively contributed to the perceived-as-genuine expressions, irrespective of the type of facial expressions. These findings provide the first evidence for dynamic facial cues to the perceived genuineness of happiness and anger expressions.

## Introduction

People with perceived-as-genuine smiles are often judged as being more attractive, friendly, and trustworthy than those who show perceived-as-deliberate smiles, thereby eliciting cooperative behaviors from decoders [1]. In contrast, perceived-as-genuine angry expressions read from a sport's team coach may cause players to cower while playing their sport [2]. Given the endogenous nature of perceived genuineness posited to increase the trustworthiness of the expresser by communicating the need to embark upon and ensure successful social interaction [3], perceived-as-genuine expressions can be expected to have more significant impacts on

K.N., Grant-in-Aid for Scientific Research on Innovative Area (17H06344) from JSPS, and by Moonshot R&D (JPMJMS2012) from JST to K.W. The funders had no role in study design, data collection and analysis, decision to publish, or preparation of the manuscript.

**Competing interests:** The authors have declared that no competing interests exist.

decoders' behavior when compared with perceived-as-deliberate expressions. Indeed, Krumhuber et al. [4] revealed that perceived-as-genuine smiling interviewees were more likely selected for the simulated job. Recent studies also have demonstrated that perceived-as-genuine expressions, more than perceived-as-deliberate ones, make decoders behave pro-socially in several experimental settings [3–6].

It remains unclear, however, what facial morphological features and spatio-temporal dynamics drive the perceived genuineness of facial expressions. For the morphological aspects of genuine facial expressions, the Duchenne smile has been described as one of the most famous representatives of the genuine expression [7]. The Duchenne smile is defined as a smile that involves the activation of the orbicularis oculi muscle (raising the cheek), and it is known that the genuineness of positive emotions perceived from encoders depends on whether the cheek is raised [1]. Originally, the Duchenne smile was associated with signs of positive emotions, such as enjoyment [7–10]. However, a recent study has suggested that raising the cheek can be regarded as an artifact of smile intensity rather than an indicator of positive emotion [11]. As for the temporal aspects, genuine smiles—more than deliberate ones—had longer durations between the onsets and offsets of lip corner movements [12–15]. Perusquía-Hernández et al. [16] also reported that an electromyography-based automatic detection machine trained with the temporal dynamics of smiles was able to discriminate genuine smiles from deliberate ones. More recently, Sowden et al. [17] demonstrated, using facial landmarks, that speed of facial movements differentiates deliberate expressions of anger, happiness and sadness. On the other hand, Ambadar et al. [18] clearly acknowledged the difficulty of determining whether encoders' intended meanings agreed with those perceived by decoders. Although facial expression clues from the encoders' perspective influence perceptions and judgments of smile genuineness [4, 6, 19], decoders' perceived meanings and encoders' genuine expressions must be investigated. Considering that facial expression information depends on the decoder's interpretation, evidence that encompasses both perspectives would result in a deeper understanding of facial expressions.

Beyond the aspect of the encoder, some studies have investigated facial expressions from decoders' interpretations [20]. For example, using randomly generated facial movements in avatars and their decoders' categorizations based on specific emotions, functions, and affect grids, Jack and their colleagues found that face movements matched these categories [21–23]. Although this data-driven approach has provided outstanding findings on the spatio-temporal features of facial expressions that correspond to the decoders' interpretation, the practical constraint on the kinetic potential of facial expressions is not guaranteed from the ecological validity viewpoint as decoders have observed facial avatars rather than real human faces. Further, the particular spatio-temporal features most important in the human perception of what is genuine vs deliberate remain an open issue. To further understand the spatio-temporal features of facial expressions, it would be desirable to investigate actual human facial expression movement instead of avatars and to compensate for them.

The current study aimed to clarify the spatio-temporal features of perceived-as-genuine facial expressions by having participants judge whether real human faces show genuine or deliberate emotions on the basis of their facial movements. Dawel et al. provide genuine/false norms for facial expressions, but their analysis mainly relies on visual inspection of facial photographs without a quantitative analysis of the spatio-temporal features of facial expressions [24]. Ambadar et al. [18] also suggest that perceived-as-amused smiles consist of enhanced cheek raising, an open mouth with a larger amplitude, and a longer duration than perceived-as-polite smiles. However, there are two methodological limitations in the study. First, the number of coded facial movements is limited. Moreover, the number of video frames required to record spontaneous facial expressions differ, which makes it difficult to quantitatively

compare between perceived-as-amused and perceived-as-polite smiles. To overcome the methodological problems, the current study developed perceived-as-genuine/deliberate expressions and examined their spatio-temporal features using deliberate expressions' facial databases, in which the number of frames and position of peak are controlled. Furthermore, we tested anger expressions as well as happy expressions, whereas many other scholars have only studied happy expressions. It is important to investigate perceived-as-genuine anger because the decoders' interpretation of angry facial expressions depends on the genuine vs. deliberate axis as much as happy ones do [25].

More concretely, the participants in this study judged the genuineness of a set of dynamic facial databases of happiness and anger. Then, this study explored which spatial pattern related to the decoders' judgment of genuineness, using a mixed model that explicitly modeled encoder and decoder effects. After that, we identified the spatio-temporal features of the perceived-as-genuine and perceived-as-deliberate facial expressions of happiness and anger, using a state-space model with change point detection of spatial component changes over time [26, 27].

We anticipated that the spatial patterns of both expressions would correspond to the proto-typical expressions predicted by basic emotion theory (BET) [28]. Krumhuber et al. [29] found that deliberate expressions were more prototypical in their facial patterns than spontaneous ones. Therefore, we expected that these prototypical spatial patterns would decrease the decoders' judgment of genuineness and be enhanced in perceived-as-deliberate expressions more than in perceived-as-genuine expressions. The prototype of happiness is a smile with a contraction of the orbicularis oculi muscle, while the prototype of anger is facial movements composed of lowering the eyebrows, widening the eyes, and tightening the lower eyelids. It should be noted that this study did not aim to evaluate the validity of facial expressions based on the BET [30, 31]. Considering the previous findings for the temporal patterns from encoders [12–15], we anticipated that the onset would be faster with perceived-as-deliberate expressions than with perceived-as-genuine expressions.

## Methods

### Participants

A total of 89 crowdsourcing workers (64 women and 25 men: *age range* = 19–73, *Mean* = 37.92, *SD* = 10.79) agreed to participate in a survey via Crowdworks (CW: www.crowdworks.jp), and all participants were Japanese. The validation of CW participants has already been confirmed by Majima et al. [32] and is aligned with that of the normal participants of behavioral experiments. Informed consent on the CW platform was obtained from each participant before the investigation in line with a protocol approved by the Ethical Committee of the Graduate School of Education, Hiroshima University (2019086), and the Institutional Review Board of Waseda University (2015–033). This study was conducted in accordance with the ethical guidelines of our institute and the Declaration of Helsinki. After completing the experimental task, the participants received 900 JPY for completing a 60-min survey.

### Stimuli

This study used prerecorded video clips of facial expressions from 20 Japanese models (50% women: *age range* = 21–33, *mean* = 26.60, *SD* = 3.22). This dynamical facial database was developed by another research project. The models were asked to show facial expressions according to six emotions (anger, happiness, disgust, fear, sadness, and surprise) under four emotional scenarios and to show a neutral expression four times. The models were instructed to maintain a neutral expression for the initial 4 seconds and then show an intended emotion on their faces for 5 seconds in a way they thought natural. To aid the models in producing

their expressions according to the time course, the timing of initiating expressions was indicated by a pure tone (1000 Hz) produced from a speaker system, followed by sound presentation every second. This instruction aimed to show the models how to deliberately produce their facial expressions within a certain time range to make it easier to compare between expressions at the expense of the natural time course of facial expressions. All video sequences had 1920 x 1440 pixel resolutions at 30 frames per second and were targeted, ranging from −2000 ms to +2000 ms from the onset of facial movements (start of a pure tone), resulting in 121 frames (4 seconds). The current study extracted only three types of emotions (i.e., anger, happiness, and neutral), of which anger and happiness of the same two men and women were excluded, due to time constraints and human resources involved in the viewed expressions. Consequently, the current study used 16 (models) x 2 (emotion: anger, happiness) x 4 (scenarios) plus neutral expressions by 20 (models: 148 total clips). Example stories of anger and happiness are the following: "when you are blamed even though you are not at fault at all (angry1)," "when someone insults your family (angry2)," "when you enjoy conversation with your friends (happy1)," and "when someone praises you (happy2)."

## Procedure

This study used the Gorilla Experiment Builder (www.gorilla.sc) to create and host our experiment [33]. Data were collected between November 29, 2019, and December 27, 2019. All participants were asked to provide consent via a check-box if they wished to participate. Thus, written type of consent was informed and obtained. This form of consent was approved by the Ethical Committee of the Graduate School of Education, Hiroshima University. This was the only form of consent that was given. On the experimental platform, the participants provided some basic demographic information (age and sex). After this, they were given careful instructions about the concept of genuine and deliberate facial expressions and their requirements as participants, followed by Namba et al. [34]. The following instruction was given in Japanese: "People sometimes express genuine facial expressions caused by actual emotional experiences, while some people can express deliberate facial expressions of emotion by intentional manipulation. In this study, we aim to understand whether people have the ability to detect whether or not the person depicted is feeling each emotion." Unknown to the participants, all expressions were deliberate. Next, all the participants performed practice trials with two facial stimuli not used in the main trials (two intended smiles expressed by the experimenter). When the participants completed the practice trials, the platform confirmed that the participants understood the task. If the participants responded with no questions, the main trials began. However, if there were issues understanding the task, the participants were reminded of the instructions and asked to redo the practice trial. The main task program presented expressions from a pool of 148 dynamic facial stimuli. We asked participants to judge whether the target person expressed genuine or deliberate expressions. The order of facial stimuli was randomized. All clips were played once, and the inter-stimulus interval was exactly 300 ms.

Following the main task, the participants filled out the Japanese version of four questionnaires related to social cognition: the Social Interaction Anxiety Scale [35, 36], the Social Phobia Scale [35, 36], the Emotional Contagion Scale [37, 38], and the Interpersonal Reactivity Index [39]. These metrics were measured for another relevant research project [40] on emotional perception, and thus we did not report the results using these questionnaires.

## Statistical analysis

To the happy (N = 64) and angry (N = 64) facial stimuli, we extracted frame-level action unit (AU) intensities on a 5-point scale with an automatic AU detection system (Openface [41,

42]). The Facial Action Coding System considers AUs as having the ability to describe all facial movements anatomically [28]. While OpenFace does not guarantee the same performance that manual facial coding does, there was sufficient biserial correlation ($r = .80$) between OpenFace and expert FACS coders' performances to static frontal facial images of Japanese persons [43]. OpenFace can detect 18 AUs: 1 (inner brow raiser), 2 (outer brow raiser), 4 (brow lowerer), 5 (upper lid raiser), 6 (cheek raiser), 7 (lid tightener), 9 (nose wrinkler), 10 (upper lip raiser), 12 (lip corner puller), 14 (dimpler), 15 (lip corner depressor), 17 (chin raiser), 20 (lip stretcher), 23 (lip tightener), 25 (lips parts), 26 (jaw drop), 28 (lip suck), and 45 (blink).

To reduce the dimensionality and extract the low-dimensional features, a nonnegative matrix factorization was applied to the time-series data of the AUs [44–46]. This approach helps obtain interpretable features in a low-dimensional space [44]. Indeed, the nonnegative matrix factorization [47] is the space-by-time manifold algorithm and is suitable for identifying the dynamic facial patterns that extract spatial (AU combination) patterns with reduced dimensions and time-series changes [48, 49]. Chiovetto et al [50] also permitted very low-dimensional parametrization of the associated facial expression with emotion, using a similar approach. The factorization rank was determined by the cophenetic coefficients [51].

To clarify the relationships between identified NMF patterns and decoders' dichotomous judgments of them as genuine or deliberate, a generalized linear mixed model was conducted to control for the differences between each encoder and decoder. In addition, we adopted a Bayesian approach to evaluate uncertainty as probability distributions. The models in this study are described as follows:

$$Y_{judgement\ of\ genuiness} \sim Bernoulli(p)$$

$$logit\ p = \beta_{intercept} + \beta_1 * Component1 + \beta_2 * Component2 + \beta_3 * Component3$$

$$\beta_{intercept} = \gamma_{intercept} + \gamma_{encoder} + \gamma_{decoder}$$

All predictors were standardized to improve the interpretation of the coefficients. All priors were kept at the default settings for the brm function [52]. If the 95% credible interval of the parameters does not include zero, a significant effect could be inferred to have been identified.

Based on the decoders' dichotomous judgments of the presented expression as genuine or deliberate, we divided facial expressions into the following three types: the relatively perceived-as-genuine, the ambiguous, and the relatively perceived-as-deliberate facial expressions. Of the happy/angry facial stimuli, we extracted the +0.8/+1.0 SD adjudged genuine, as well as the −0.8/−1.0 SD stimuli adjudged deliberate (Fig 1). Finally, the number of target facial expressions was 128 (16—eleven women and five men—perceived-as-genuine happiness; 36—eighteen women and men—ambiguous happiness; 12—three women and nine men—perceived-as-deliberate happiness; 13—ten women and three men—perceived-as-genuine anger; 37—nineteen women and eighteen men—ambiguous anger; 14—three women and eleven men—perceived-as-deliberate anger). Taking each frame (121) in each video resulted in 15,488 data points (121 frames x 128 expressions). These expressions were employed to systematically generate facial expressions considered perceived-as-genuine/ambiguous/deliberate expressions and not the same as participants' to estimate population indices for effect sizes. Consequently, power analyses were not available. The N of 64 for each emotion was chosen as more than the usual number of expressers employed in the research using the actor's facial expressions, which was likely to produce stable means and allow for conducting multivariate statistical analyses [53]. Moreover, this sample size is expected to emphasize the more distinctive descriptions of each perceived-as expression.

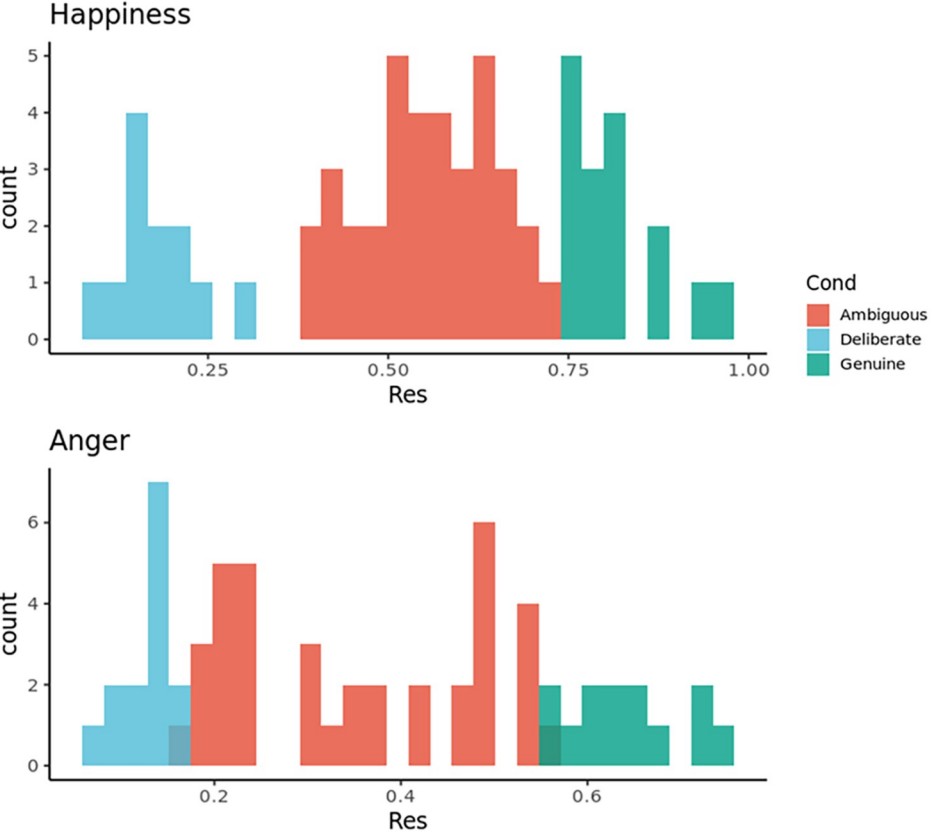

**Fig 1. The histogram of Yes responses for the genuineness judgment of happiness (upper part) and anger (lower part).**

For the temporal features, we applied a state-space model with the change point detection to spatial component changes over time [26, 27]. The model can be described as follows:

$$\mu[t] \sim Normal(2\mu[t-1] - \mu[t-2], \sigma_\mu)$$

$$\delta1[t] \sim Cauchy(\delta1[t-1], \sigma_{\delta1})$$

$$\delta2[t] \sim Cauchy(\delta2[t-1], \sigma_{\delta2})$$

$$Y_{ambiguous}[t] \sim Normal(\mu[t], \sigma_Y)$$

$$Y_{genuine}[t] \sim Normal(\mu[t] + \delta1[t], \sigma_Y)$$

$$Y_{deliberate}[t] \sim Normal(\mu[t] + \delta2[t], \sigma_Y)$$

where $Y$ are the observable matrices of the spatial component matrix, and $t$ means the frame or time. $\mu$ is the spatial component matrix common to three expressions. $\delta1$ / $\delta2$ can be considered the magnitude of difference between the perceived-as-genuine/deliberate/ambiguous expressions. A prior distribution without any specification is a uniform distribution. The code is available on Open Science Framework (OSF: https://osf.io/e7pdt). If the $\delta$ terms are greater

than zero (i.e., positive value), this means that the spatial component of perceived-as-genuine/deliberate is relatively large, and if it is smaller than zero (i.e., negative value), this means that the spatial component of perceived-as-genuine/deliberate expressions is relatively smaller than that of ambiguous expressions. We calculated the 99% credible interval of the $\delta$ as to whether the intervals fall to zero could be considered as the testing for $\delta$.

To develop the spatio-temporal patterns from AU data, we used the "NMF" packages [54] in R to implement the calculation. As for the generalized linear mixed model, all iterations were set to 3,000 and burn-in samples were set to 1,000, with the number of chains set to four using the "brms" package [52]. For a state-space model, we used the "cmdstanr" package [55] and set all iterations to 15,000, as well as burn-in samples to 5000. The value of R-hat for all parameters equaled about 1.0, indicating convergence across the four chains [56].

## Results

### Happiness

Fig 2 shows the spatial components from all facial expressions of happiness. Visually inspecting the relative contribution of each AU to the independent components, we interpreted Component 1 as opening the mouth (AU25, 26). The results of Component 2 indicated smiling (AU12) with eye constriction (AU6, 7) and opening the mouth (AU25), while those of Component 3 suggested that raising the chin (i.e., AU17) was a main contributor. Although Component 2 also included upper lip raising (AU10) and dimpling (AU14), these AUs can be

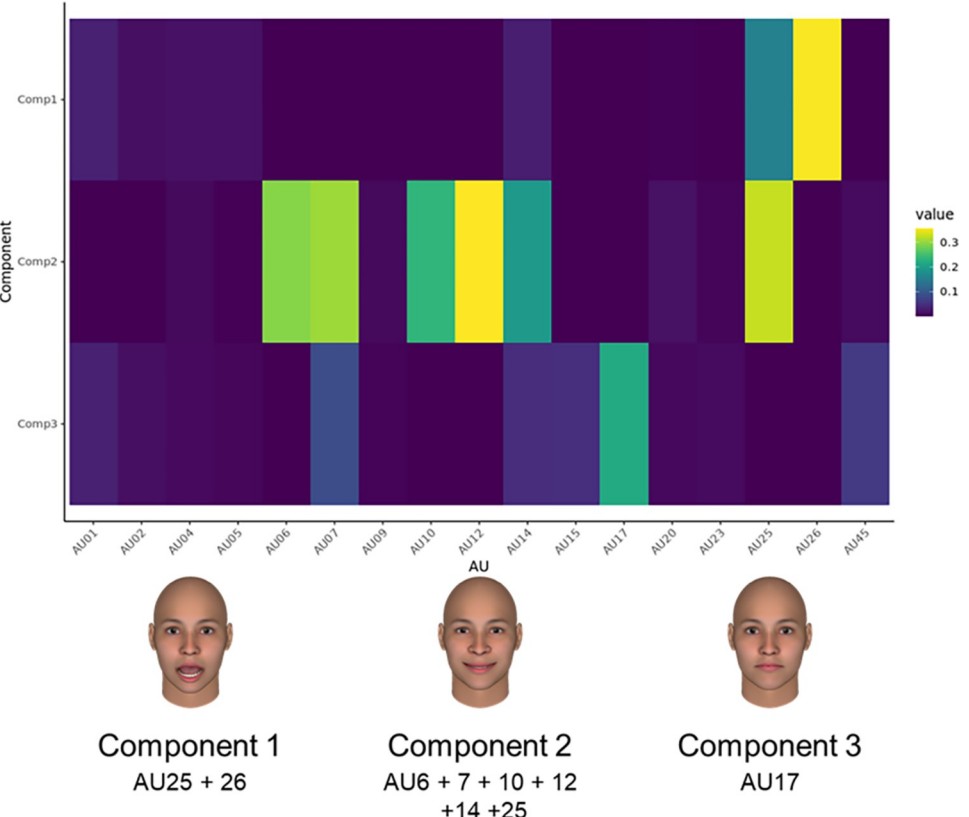

**Fig 2. Heatmap of each component's loadings for facial expressions of happiness (upper part) and visual representations (lower part). Value colors represent each facial movement's contribution to component scores.**

**Table 1. Results of the generalized linear mixed model for the relationships between identified NMF patterns and decoders' dichotomous judgments of genuineness.**

|  | Happiness | Anger |
|---|---|---|
| Random effects | Variance [95%CI] | |
| Decoders (intercept) | 0.89 [0.75, 1.06] | 1.21 [1.01, 1.44] |
| Encoders (intercept) | 1.00 [0.82, 1.22] | 0.88 [0.72, 1.08] |
| Fixed effects | EAP [95%CI] | |
| Component 1 | -0.78 [-1.07, -0.50] | -0.62 [-0.85, -0.38] |
| Component 2 | -0.46 [-0.75, -0.18] | -0.23 [-0.48, 0.01] |
| Component 3 | 0.10 [-0.17, 0.37] | -0.39 [-0.64, -0.15] |

interpreted as the confusion of AU12 in the automated action coding detection system [46, 57].

To clarify the relationships between identified NMF patterns and decoders' dichotomous judgments of them as genuine or deliberate, a generalized linear mixed model with random intercepts was built and tested to control for the differences between each encoder and decoder. Table 1 depicts the coefficients for each factor of NMF predicting genuineness judgment. Notably, Component 1 (opening the mouth) and Component 2 (smiling with eye contraction) were found to predict genuineness judgment ($\beta_1 = -0.78$, 95% Credible Intervals [$-1.07$, $-0.50$]; $\beta_2 = -0.46$, 95% CI [$-0.75$, $-0.18$]), but Component 3 (raising the chin) did not because of the 95% CI that included 0 ($\beta_3 = 0.10$, 95% CI [$-0.17$, 0.37]).

To differentiate perceived-as-genuine and perceived-as-deliberate facial expressions of happiness, Fig 3 shows the quantitative indices of the time-series patterns for the magnitude of difference between the perceived-as-genuine, ambiguous and perceived-as-deliberate expressions of happiness. S1 Table represents the 99% credible intervals and probability of directions [58, 59] at 500 ms intervals. Visual inspection of Component 1 (opening the mouth) revealed that

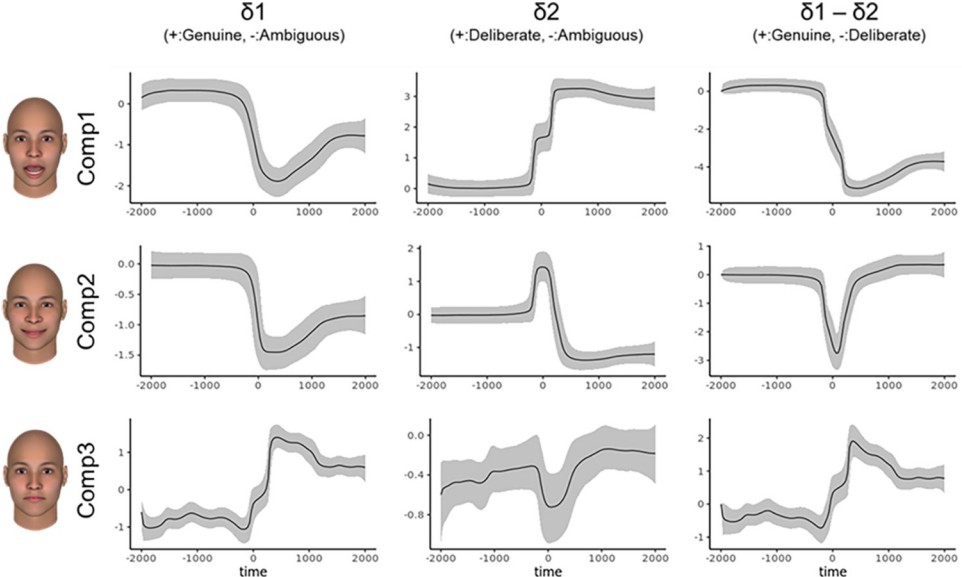

**Fig 3. Time-series patterns for the magnitude of difference between the perceived-as-genuine and perceived-as-deliberate expressions of happiness.** The y-axis represents the extent of the "δ" parameters for each component. Solid lines indicate the expected a posteriori. Positive values refer to a relatively large spatial component of (left: perceived-as-genuine, center: deliberate, right: genuine), while negative values indicate a relatively large spatial component of (left and center: perceived-as-ambiguous, right: deliberate). The ribbons represent 99% credible intervals.

the perceived-as-deliberate expressions showed a larger mouth opening, while the perceived-as-genuine expressions remained deactivated when compared with ambiguous expressions. As for Component 2 (smiling with eye contraction), the perceived-as-deliberate expressions produced more rapid facial changes than the perceived-as-genuine expressions. At the middle row in the right-hand-side column of Fig 3, the difference parameter (i.e., δ1 - δ2) clearly indicated that the perceived-as-deliberate expressions reached their peaks earlier than the perceived-as-genuine expressions did. Unexpectedly, ambiguous expressions showed a stronger smiling component as offset areas (after peak: 501–2000 ms) than the other two expressions did. Component 3 (raising the chin) can be interpreted as a byproduct of Component 1 because it corresponds to raising the chin, which also means the movement of closing the mouth.

## Anger

Fig 4 shows the spatial components from all facial expressions of anger. A visual inspection of Fig 4 shows that Component 1 was contributed to by tightening the eyelids (AU7), opening the mouth (AU25), lowering the brows (AU4), and slightly raising the upper lip (AU10). Component 2 was related to opening the mouth (AU25, 26) and lowering the brows (AU4). The results of Component 3 correspond to raising the chin (AU17).

A generalized linear mixed model with random intercepts showed the coefficients for each factor of NMF predicting genuineness judgment (Table 1). All Components were found to

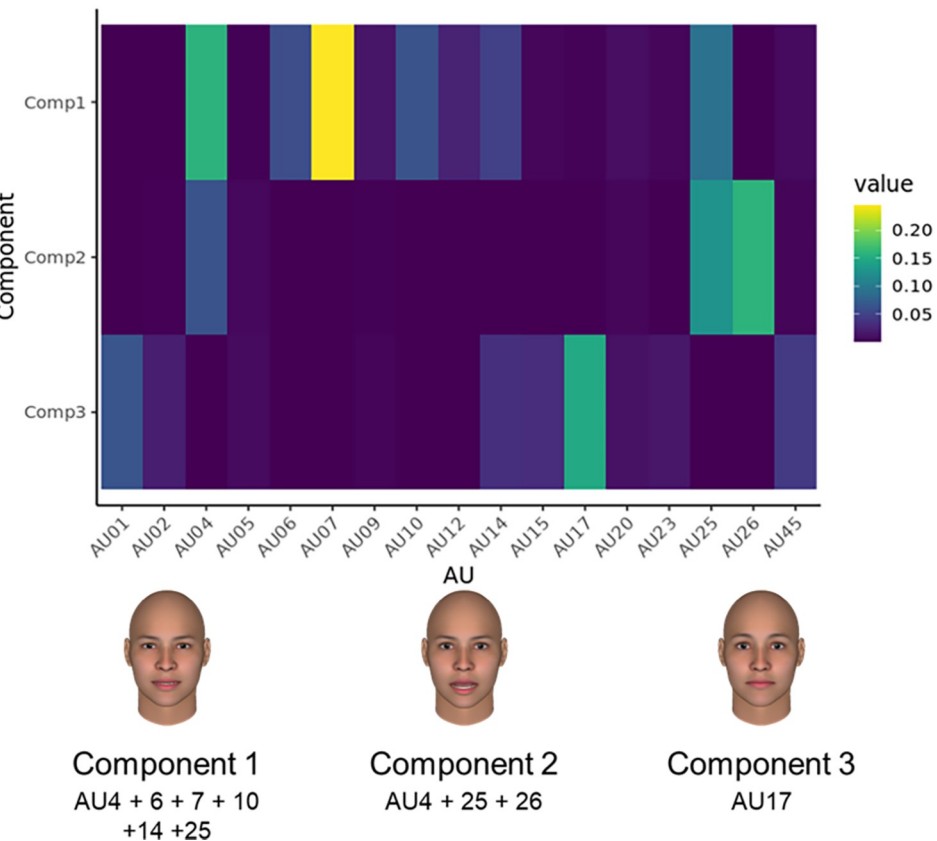

**Fig 4. Heatmap of each component's loadings for facial expressions of anger (upper part) and visual representations (lower part). Value colors represent each facial movement's contribution to component scores.**

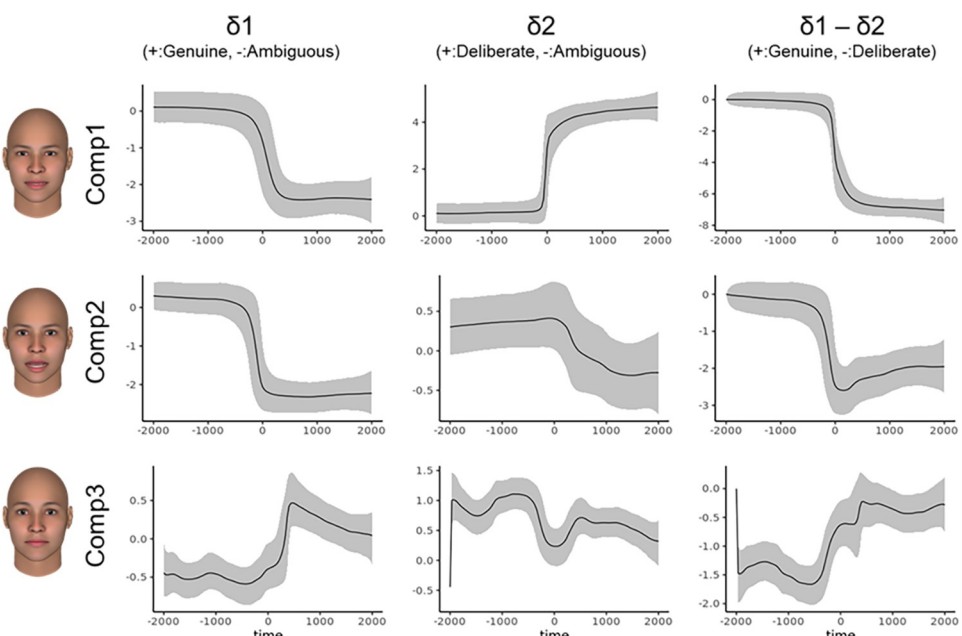

**Fig 5. Time-series patterns for the magnitude of difference between the perceived-as-genuine and perceived-as-deliberate expressions of anger.** The y-axis represents the extent of the "δ" parameter for each component. The solid lines indicate the expected a posteriori. Positive values refer to a relatively large spatial component of (left: perceived-as-genuine, center: deliberate, right: genuine), while negative values indicate a relatively large spatial component of (left and center: perceived-as-ambiguous, right: deliberate). The ribbons represent 99% credible intervals.

predict genuineness judgment ($\beta_1 = -0.62$, 95% Credible Intervals [−0.86, −0.48]; $\beta_3 = -0.39$, 95% CI [−0.64, −0.15]), but 95% CI on only Component 2 (opening the mouth) included zero slightly ($\beta_2 = -0.23$, 95% credible intervals [−0.48, 0.01]).

To differentiate the perceived-as-genuine and perceived-as-deliberate facial expressions of anger, Fig 5 indicates the quantitative indices of the time-series patterns for the magnitude of difference between the perceived-as-genuine, ambiguous, and perceived-as-deliberate expressions of anger. S2 Table represents 99% credible intervals and probability of directions at 500 ms intervals. The perceived-as-deliberate expressions contributed to Component 1, which can be regarded as multiple facial movements more so than the ambiguous and perceived-as-genuine expressions. Moreover, the perceived-as-genuine expressions showed less Component 1 (multiple frown) than the ambiguous expressions did. Component 2 (opening the mouth) had a larger peak in the perceived-as-deliberate and ambiguous expression than it did in the perceived-as-genuine expression. Component 3 (raising the chin) can be interpreted as the byproduct of Component 2 because it corresponds to raising the chin, which also indicates the movement of closing the mouth. As shown in S2 Table, there were differences between perceived-as-genuine vs. ambiguous but not perceived-as-deliberate vs. ambiguous in Component 2 after peak (0–2000 ms).

## Discussion

The current study explored the relationships between the spatial patterns of facial expressions and decoders' dichotomous judgments of them as genuine and clarified the spatio-temporal features of perceived-as-genuine and perceived-as-deliberate facial expressions. We anticipated that perceived-as-deliberate expressions would show spatial patterns typical of facial expressions and more rapid movements than perceived-as-genuine expressions. The results

produced four key findings for the spatio-temporal features of perceived-as-genuine/deliberate expressions of happiness and anger. First, some prototypical facial movements were observed for both emotions. For the happiness expression, the prototypical spatial pattern (Component 2: AU6/7 = the movement of orbicularis oculi, AU12 = the movement of the zygomatic major muscle) was observed in both the perceived-as genuine and deliberate expressions. As for the anger expression, lowering the eyebrows and opening the mouth (Component 2: AU4 = corrugator muscle, AU25 = orbicularis oris) were seen in both the perceived-as genuine and deliberate expressions, while the perceived-as-deliberate expression of anger produced several additional facial movements, including prototypical patterns (Component 1: AU4, AU7, AU25). Second, genuineness judgments were reduced when more spatial patterns were observed in facial expressions. More concretely, anger expressions included more multiple frowning (Component 1), opening the mouth (Component 2), and raising the chin (Component 3) and were perceived-as-deliberate, while happiness expressions included more opening the mouth (Component 1) and smiling with eye contraction (Component 2) and were perceived-as-deliberate. Third, the main component of happiness (Component 1) revealed that the perceived-as-deliberate expressions reached their peaks earlier than the perceived-as-genuine expressions. Finally, the movement of opening the mouth in both emotions contributed largely to decoders' dichotomous judgments of them as deliberate and the perceived-as-deliberate expressions, and the component on AU17 can be considered a byproduct of this. However, the results for opening the mouth were slightly different between happiness and anger, and in anger, the difference was remarkable with the perceived-as-genuine expressions, but the difference between ambiguous and perceived-as-deliberate ones was small. Regarding happiness, the perceived-as-genuine expressions had a small mouth opening, and the perceived-as-deliberate expressions had a large mouth opening.

Importantly, the spatial patterns inherent to prototypicality vary between emotions. As can be seen from Component 2 in Fig 3, the smiles of the perceived-as-genuine and deliberate expressions were similar in their intensity at offset (i.e., at 500–2000 ms after peak), although that of the perceived-as-deliberate expression had relatively abrupt onsets. The smile-related component in both expressions was similar, at least with respect to the final frame, and the difference in genuine/deliberate judgments might be attributable to their temporal features. The result that this spatial pattern influenced the judgment of genuineness (Table 1) also supported the contention that this temporal information is important for perceived-as-deliberate expressions. On the other hand, for anger, lid tightening (AU7), which is a part of the prototypical expressions [28] and mainly contributed to Component 1, showed significant differences between the perceived-as-genuine and deliberate expressions (Table 1 and Fig 5). The results indicate that the perceived-as-deliberate expressions consist of multiple facial actions. Fig 5 confirms that the relationship increases linearly as the degree of perceived-as-deliberate increases. By placing an ambiguous expression as an intermediate term, the current study increased the generalizability of the results. This view, that perceived-as-genuine expressions have fewer multiple frowns, is consistent with recent findings showing that deliberate anger expressions contained various facial movements more than genuine anger expressions in Asian populations [60]. The results raise the possibility that we adapt ourselves to show genuine anger expressions with fewer movements through our experiences, which might affect the judgments in the current study as well.

For the temporal aspects, as shown by Component 2 of the happiness expressions (i.e., smile-related movements shown in Fig 3), the perceived-as-deliberate expressions contained more rapid onsets than the perceived-as-genuine expressions. This result is consistent with previous findings on decoder-based facial cues [18, 61], and it can be concluded that the temporal change of perceived-as-genuine expressions should be slow when compared to the

perceived-as-deliberate ones. The indication of Sowden et al [17] that the speed of mouth-widening actions helps differentiate between happy and other emotional expressions for deliberate expressions is consistent with previous findings regarding the encoder aspects. As Fig 5 shows, with regard to anger, there were more rapid and intense onsets in Components 1 and 2 relative to the perceived-as-genuine expressions. The greater the speed the greater the perceived intensity of anger expressions [17], but rapid speeds are not always understood to be natural as found in recent android research [62]. In line with the accumulated evidence, many scholars have already reported that the temporal aspects of facial expressions are important [63–66]. Nevertheless, future studies should bear in mind that the credibility of messages on facial expressions may differ depending on the speed of their expressions.

More interestingly, the movement of opening the mouth in both emotions contributes strongly to the perceived-as-deliberate expressions. Indeed, Namba et al. [67] found a sequence emphasizing the movement to open the mouth in deliberate smiles and Sowden et al. [17] indicated that the high speed of mouth opening was important for posed expressions of happiness. The results provide the first evidence that exaggerated facial expressions, including opening the mouth, are judged to be deliberate and that this can be extended to anger as well as happiness. Especially in perceived-as-genuine (not deliberate) anger, the degree to which the mouth opens becomes smaller. However, Ambadar et al. [18] indicate the opposite results that perceived-as-amused smiles include opening the mouth more often than perceived-as-polite smiles. One possible explanation for this discrepancy is provided by the nature of the target facial database. Ambadar et al. [18] used the smiles that were not performed in response to a request, whereas the current study applied all facial expressions performed under emotional stories with express intentions. In other words, the former's spontaneous smile with high intensity differs from the latter's emphasized deliberate smile in that the cause to express and the uncontrolled duration of the expression may influence the interpretation of the intensity of the mouth opening. An alternative explanation is based on cultural differences. Since the target population of the current study was East Asians, who are prone to high context communications [68], Fang et al. [60] also reported that facial expressions are less distinct in Eastern people than in Western people. Jack et al. [69] support this because they revealed that Westerners showed their mental representations of basic emotions with more distinct facial movements when compared to Easterners. The perceived-as-genuine expressions may have been less intense and more ambiguous in terms of opening the mouth, with a context preferentially processed.

The finding for the spatio-temporal features of perceived-as-genuine and deliberate expressions might contribute to a pragmatic understanding of our emotional communication. Many researchers emphasize actual usage for facial expressions of emotion [70–72], but this remains insufficient for how it is actually expressed in daily life. Given that perceived-as-genuine facial expressions sometimes prompted the decoder to behave to the encoders' advantage [3–6], the spatio-temporal features of perceived-as expressions should induce important suggestion for future work. For example, in android research, this finding, that lower degrees of opening the mouth and prototypical components enhances genuineness, may contribute to the development of more elaborate "emotional" robots, which can be considered perceived-as-genuine. We will need to continue our efforts to acknowledge and describe the complexity of our emotional communication.

Notably, unexpected gender differences were observed in perceived-as-genuine expressions, that is, more female faces were included in perceived-as-genuine expressions, while more male faces were included in perceived-as-deliberate expressions. This might be partly attributed to the higher perceived emotionality, honesty, and trustworthiness often associated with female-appearing facial features [73, 74], which leads to the perceptual bias that female

actors show genuine expressions more frequently than male ones. The current study also included more female than male perceivers, which suggests that the gender imbalance in the pool was due to the random collection of CW data. However, as Spies and Sevincer [75] argued, women tend to be more accurate in distinguishing between authentic and nonauthentic smiles, which is consistent with the study's purpose that is to examine perceived-as-genuine facial expressions compensating for encoders' genuine expressions.

While the current study showed the spatio-temporal features of perceived-as-genuine and deliberate expressions, there are limitations to be noted here. First, all facial expressions were essentially deliberate by following emotional stories. If genuine expressions have specific associated movements (e.g., [45]), the current facial database cannot be used to identify them. Therefore, future studies would benefit from accumulating empirical findings from human/avatar facial expressions and encoder/decoder perspectives. While the current study used all deliberate human expressions at the expense of ecological validity, this methodology has an advantage in controlling the overall duration and the position of the peak. Previous studies point out that there may be multiple peaks in spontaneous facial reactions [49, 76], and thus, future research will need to take into account such complexity that cannot be investigated in deliberate expressions. Further, 2000 ms before and after the peak of expression were arbitrarily extracted in this study. It has been reported that offset is important for decoders [77]. It is important to consider including complete ranging in offset as opposed to onset when using the other deliberate expression database. Second, the results of this study are only based on Japanese samples. Rychlowska et al. [78] have argued that historical heterogeneity is associated with norms favoring greater emotional expressivity. Niedenthal et al. [79] suggest that historically heterogeneous societies promote expressivity and clarity in emotional expressions. Given that Japan has populations of historically homogeneous societies that share common values and rely on more indirect and ambiguous communication depending on contextual information [80], the finding of the current study can be culture specific. It should also be noted that the experiments could not be controlled well as they were conducted online and several studies have suggested that crowd worker data sometimes do not achieve reliable quality [81]. Therefore, it will be necessary to consider such cross-cultural perspectives in future studies that use laboratory experiments or more online experiments that include attention-check questions. Third, forcing yes-or-no responses from decoders throws away valuable information about the degree of perceived genuineness [82]. Although using the extreme group analysis that the current study applied (i.e., the most perceived-as expressions) has been justified by a simulation study [82], it would be desirable to use a rating scale for authenticity instead of a yes-or-no response because the rating scale's perceived genuineness of different stimuli is expected to provide much more information [24].

Finally, the current automated evaluation system of the AU can provide several AU intensities at a frame-by-frame level. This is an advantage of using the automated AU detection system; however, it is not perfect despite recent developments in machine learning and artificial intelligence techniques in the area of affective computing [83]. Indeed, for Component 3, the differences between the perceived-as-genuine/deliberate and ambiguous expressions were often observed before the peak frame (Figs 3 and 5). This may reflect noise that is a fit to the individual's face morphology rather than to facial expressions of emotion. It should be noted that the assessment of facial movements is largely dependent on the target stimuli and their nature [84], but the state-of-the-art AU detection system comparisons provided average F1 scores of .56–.59 [85]. Perusquia-Hernández et al. [46] also indicate the existence of entanglement between upper lip raising (AU10) and lip corner pulling (AU12). Replication studies with a more sophisticated facial movement detection system are awaited.

To summarize, the current study revealed the spatio-temporal features of the perceived-as-genuine and deliberate facial expressions of happiness and anger. In the case of the happiness expression, the smile-related spatial pattern occurred in both perceived-as expressions. For the anger expression, lowering the eyebrows and opening the mouth were seen in both expressions, but the perceived-as-deliberate expression produced multiple facial movements, including squeezing the eyes. In addition, the perceived-as-deliberate expressions had a faster onset to the peak than the perceived-as-genuine expressions. Less movement of opening the mouth in both emotions contributes strongly to the perceived-as-genuine expressions. Identifying the spatio-temporal features of the perceived-as-genuine expressions can contribute to building facial databases that can evoke decoders' reactions based on the credibility of the nonverbal message. Moreover, it may enrich the affective computing areas by applying to humanoid robots that purport to express human-like displays.

## Supporting information

**S1 Table. Results for the magnitude of difference between the perceived-as-genuine and perceived-as-posed expressions of happiness compared to ambiguous expressions.** (XLSX)

**S2 Table. Results for the magnitude of difference between the perceived-as-genuine and perceived-as-posed expressions of anger compared to ambiguous expressions.** (XLSX)

## Author Contributions

**Data curation:** Shushi Namba.

**Formal analysis:** Shushi Namba.

**Funding acquisition:** Katsumi Watanabe.

**Investigation:** Shushi Namba.

**Methodology:** Shushi Namba.

**Project administration:** Katsumi Watanabe.

**Supervision:** Koyo Nakamura.

**Validation:** Koyo Nakamura, Katsumi Watanabe.

**Writing – original draft:** Shushi Namba.

**Writing – review & editing:** Koyo Nakamura, Katsumi Watanabe.

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
