## [Decision Letter · Decision Letter 0]

13 Apr 2022

PONE-D-21-33666The spatio-temporal features of perceived-as-genuine and deliberate expressionsPLOS ONE

Dear Dr. Namba,

Thank you for submitting your manuscript to PLOS ONE. After careful consideration, we feel that it has merit but does not fully meet PLOS ONE’s publication criteria as it currently stands. Therefore, we invite you to submit a revised version of the manuscript that addresses the points raised during the review process.

I have now received two reviews of the manuscript entitled The spatio-temporal features of perceived-as-genuine and deliberate expressions (PONE-D-21-33666) that you submitted to PLOS ONE. I was fortunate to secure reviews from two experts in the areas of facial expression analysis and emotional genuineness, both of whom provided strong insight to the strengths and weaknesses of the manuscript.  I read the original manuscript, and then again with comments provided by the reviewers. As you will see, the reviewers found the manuscript to be interesting, and the analytic techniques impressive. The topic is also relevant to the broad readership at PLOS One.  Both reviewers identified the academic merit of the manuscript, however noting that the clarity of the manuscript could be improved. Both reviewers provided guidance on improving the readability, which I encourage you to consider. There was disagreement as to the statistical & methodological rigor of the work, with Reviewer 1 taking issue with some analysis decisions. Both suggested major revisions. As such, I am rejecting the manuscript in its current form, and I invite you to resubmit in the form of major revisions.  You will find the full set of reviewer comments included below. To these I add my own two comments:  1) I would appreciate seeing a clear statement of theoretical contribution of this work. I read your 2021 Scientific Reports article with great interest. However, the 2021 work is only minimally reviewed, and readers may be left unclear as to what this work contributes to our understanding of genuine expressions. I recognize that there are differences with this work – OpenFace vs DeepLabCuts, Happy-Angry vs Surprise, ActionUnits vs landmarks, but these differences are methodological, rather than conceptual. I believe the core contribution relates to the perceptual nature of genuine expressions, but this should be more clearly contrasted from your previous works, along with those in the field. 2) The stimuli presenting Genuine-vs-Deliberate expressions are central to the work, with both perceptual and motion analyses relying on their validity. Yet there is almost no discussion as to how these stimuli were produced, and whether the scenarios would be considered ‘genuine’. This presents a serious methodological oversight which would need to be robustly addressed in your revision.

We look forward to receiving your revised manuscript.

Kind regards,

Steven R. Livingstone

Academic Editor

PLOS ONE

2. Please change "female” or "male" to "woman” or "man" as appropriate, when used as a noun (see for instance https://apastyle.apa.org/style-grammar-guidelines/bias-free-language/gender).

“This research was supported by Early-Career Scientists (20K14256) from JSPS to S. N., Early-Career Scientists (19K20387) from JSPS to K.N., Grant-in-Aid for Scientific Research on Innovative Area (17H06344) from JSPS, and by Moonshot R&D (JPMJMS2012) from JST to K.W.”

“This research was supported by Early-Career Scientists (20K14256) from JSPS to S. N., Early-Career Scientists (19K20387) from JSPS to K.N., Grant-in-Aid for Scientific Research on Innovative Area (17H06344) from JSPS, and by Moonshot R&D (JPMJMS2012) from JST to K.W.”

“This research was supported by Early-Career Scientists (20K14256) from JSPS to S. N., Early-Career Scientists (19K20387) from JSPS to K.N., Grant-in-Aid for Scientific Research on Innovative Area (17H06344) from JSPS, and by Moonshot R&D (JPMJMS2012) from JST to K.W.”

Additional Editor Comments:

Dear Dr. Namba,

I have now received two reviews of the manuscript entitled The spatio-temporal features of perceived-as-genuine and deliberate expressions (PONE-D-21-33666) that you submitted to PLOS ONE. I was fortunate to secure reviews from two experts in the areas of facial expression analysis and emotional genuineness, both of whom provided strong insight to the strengths and weaknesses of the manuscript.

I read the original manuscript, and then again with comments provided by the reviewers. As you will see, the reviewers found the manuscript to be interesting, and the analytic techniques impressive. The topic is also relevant to the broad readership at PLOS One.

Both reviewers identified the academic merit of the manuscript, however noting that the clarity of the manuscript could be improved. Both reviewers provided guidance on improving the readability, which I encourage you to consider. There was disagreement as to the statistical & methodological rigor of the work, with Reviewer 1 taking issue with some analysis decisions. Both suggested major revisions. As such, I am rejecting the manuscript in its current form, and I invite you to resubmit in the form of major revisions.

You will find the full set of reviewer comments included below. To these I add my own two comments:

1) I would appreciate seeing a clear statement of theoretical contribution of this work. I read your 2021 Scientific Reports article with great interest. However, the 2021 work is only minimally reviewed, and readers may be left unclear as to what this work contributes to our understanding of genuine expressions. I recognize that there are differences with this work – OpenFace vs DeepLabCuts, Happy-Angry vs Surprise, ActionUnits vs landmarks, but these differences are methodological, rather than conceptual. I believe the core contribution relates to the perceptual nature of genuine expressions, but this should be more clearly contrasted from your previous works, along with those in the field.

2) The stimuli presenting Genuine-vs-Deliberate expressions are central to the work, with both perceptual and motion analyses relying on their validity. Yet there is almost no discussion as to how these stimuli were produced, and whether the scenarios would be considered ‘genuine’. This presents a serious methodological oversight which would need to be robustly addressed in your revision.

Reviewers' comments:

Reviewer's Responses to Questions

**Comments to the Author**

1. Is the manuscript technically sound, and do the data support the conclusions?

Reviewer #1: Partly

Reviewer #2: Yes

2. Has the statistical analysis been performed appropriately and rigorously? 

Reviewer #1: No

Reviewer #2: Yes

3. Have the authors made all data underlying the findings in their manuscript fully available?

Reviewer #1: Yes

Reviewer #2: Yes

4. Is the manuscript presented in an intelligible fashion and written in standard English?

Reviewer #1: No

Reviewer #2: Yes

5. Review Comments to the Author

Reviewer #1: PRAISE

This manuscript explored the interesting question of whether there are morphological and temporal differences between facial behaviors that are perceived to be genuine vs. deliberate. I liked that it examined perceptions of human facial behaviors rather than computer-generated facial behaviors and agree that this is an important validation step. I also liked the attempt to simultaneously consider morphology and temporal characteristics of the behavioral displays. I also thought the visualizations were nice and appreciated the data and materials being made open.

CONCERNS

1. The manuscript’s readability would be helped if the authors were more consistent in their use of terms. As just one example, multiple different terms are used to refer to the person producing a facial behavior (e.g., sender, encoder) or perceiving it (e.g., observer, decoder, perceiver).

2. The methodological decision to look at onset plus and minus 2 sec creates some standardization but also erodes some of the ecological validity of the stimuli. If temporal characteristics matter for perception, as the authors contend, then why cut them partway through and therefore eliminate the ability of perceivers to see their duration, offset, and other temporal features?

3. Please provide a rationale for making the genuineness vs. deliberate ratings dichotomous. Why not have them rate this quality on a continuum? Forcing a choice between two alternatives throws away valuable information about degree (DeCoster et al., 2009) and introduces (or at least exacerbates) participants’ subjective thresholds as a source of variability.

4. The first paragraph in the Statistical Analysis section (page 6) was very difficult to understand. I think the authors calculated the percentage of participants that rated each stimulus as genuine and then used that quantity to select a subset of the stimuli most consistently rated as genuine vs. deliberate. But the descriptions and some words (“prescribed” and “judgment ratios”) were not adequately clear/explained.

5. Related to concern #4 above, I did not love the methodological decision to analyze 40 stimuli rather than all 180. I think this was done in order to compare behaviors that are perceived-by-most-as-genuine to perceived-by-most-as-deliberate rather than having more ambiguous behaviors. But this selection makes the results much less generalizable. We are not learning about most facial behaviors, we are learning about the 22% least ambiguous ones. If genuineness had been rated dimensionally (as in concern #3), the authors could have regressed it on various temporal features and included all 180 stimuli. If they used a mixed effects model, they also could have explicitly modeled encoder and decoder effects.

6. The authors acknowledge that OpenFace may be incorrect but do not provide any validation evidence such as comparing it to expert FACS coding on a subset of this specific dataset. Because we don’t know how much to trust the AU intensity estimates, we don’t know how much to trust the NMF components and these are really important variables in the paper.

7. Please add some information to the Statistical Analysis section about you will be calculating 99% credible intervals at each value of t and how to interpret the figures (e.g., seeing whether delta=0 falls within that interval). Some of this information is currently in the figure captions only and it is very easy to miss.

8. In addition to plotting the 99% credible intervals at each value of t, I would be interested to see tests done on larger regions of t-space. For example, the authors could calculate the average delta score in the regions from t=0 to t=2000 in intervals of 250 or 500ms. They could also report probability of direction (pd) values for these regions in addition to credible intervals (Makowski et al., 2019).

COMMENTS

1. The claim on page 1 line 50 that perceived-as-genuine expressions have more significant impacts on observer’s behavior than perceived-as-deliberate expressions is a leap that does not follow directly from the evidence presented earlier in the paper.

2. Page 4 would be improved by providing an explicit rationale for the decision to look at anger and happiness rather than other possible emotions.

3. On page 5, please briefly describe what the four emotional scenarios were for happiness and anger.

4. Were there are quality or attention check questions to ensure that the participants were taking the task seriously? I’m not sure if this is a big issue in Japan, but in the US many crowdworkers do a poor job and this can be detected with some basic questions.

REFERENCES

DeCoster, J., Iselin, A.-M. R., & Gallucci, M. (2009). A conceptual and empirical examination of justifications for dichotomization. Psychological Methods, 14(4), 349–366. https://doi.org/10/bh86w7

Makowski, D., Ben-Shachar, M. S., Chen, S. H. A., & Lüdecke, D. (2019). Indices of effect existence and significance in the Bayesian framework. Frontiers in Psychology, 10. https://doi.org/10/ggfw2j

Reviewer #2: Thank you for inviting me to review this manuscript. The authors investigate spatio-temporal features of facial emotion expressions that help to distinguish perceived-as-genuine and perceived-as-deliberate (i.e., non-authentic) expressions. They do so by first presenting a series of dynamic facial emotion expression videos to 89 Japanese individuals and asking them to determine whether each expression is genuine or deliberate (completed for happy and angry expressions). They then reduced each expression video to its lowest dimensional features (as a function of the emotion and 'perceived as' conditions) using non-negative matrix factorisation. Both perceived-as-genuine and perceived-as-deliberate expressions showed protocol typical spatial action units, whilst deliberate expressions produced multiple facial movements in addition to these. Deliberate expressions also had faster onsets to peak expression and had a greater contribution of mouth opening movements.

The question being investigated here is novel, interesting and important. The techniques utilised to answer this question are in my opinion strong and in line with a number of recent studies applying dimensionality reduction techniques to facial emotion expressions. I commend the authors for this. I find the results very interesting and I also think they did a good job to put their findings in context as well as discuss the limitations (e.g., cultural differences that may exist and require further attention.

I have some comments on the write-up (mostly focused on the introduction) that I believe would strengthen the manuscript if rectified:

Introduction:

1) I feel the introduction could be tightened up a little bit so it really 'tells the story' for your reader to follow as to the rational for the current study. Some parts I struggled to follow the relevance of. For example, I wasn't sure I followed how the study is specifically related to the affective pragmatics theory and why this needs to be linked to an 'evolutionary account' that really had very little explanation. Perhaps you can build a story for the studies importance without these theories, especially as you don't refer back to this in the discussion, so I'm not sure exactly how relevant it is. Or alternatively, tighten up the explanation and refer back to it in the discussion.

2) I think your design is great, whereby you are able to determine what spatio-temporal features are important in the human perception or labelling of genuine vs deliberate (as opposed to determining yourselves what a genuine or deliberate expression looks like) and I agree this is in line with the work by Rachel Jack and colleagues. However, I think you could make this point and emphasise the benefit of this kind of approach clearer for the reader. You could tighten up the explanation of Jack and co.'s method.

3) My colleagues and I have investigated temporal features (mean speed) of happy, angry and sad expressions in posed and spontaneous contexts (Sowden et al., 2021; https://psycnet.apa.org/fulltext/2021-22419-001.html) which might also fit well into your introduction and/or discussion to support differences here. We found (in Experiment 1C) that speed of mouth opening and widening actions helped to differentiate between happy, angry and sad expressions only for posed and not spontaneous expressions, whilst speed of eyebrow widening movements was capable of differentiating these emotions for both posed and spontaneous expressions. This to me seems a nice finding alongside your findings, whilst your design has the nice element of looking at timecourses as opposed to our mean speed over entire expression period.

Methods

1) I think the methods section would benefit from a little more explanation of what the NMF dimensionality reduction technique is and what it is essentially doing, for readers who do not know about this technique. On this topic, the authors might be interested in (and may wish to cite) an interesting paper by Chiovetti et al. (2018; https://jov.arvojournals.org/article.aspx?articleid=2678770) who seek to reduce various facial expressions down to their lowest dimensional organisation (regarding temporal properties of expressions) whereby emotion can be determined. They also compare NMF and a Fourier-based Anechoic Demixing Algorithm in doing so.

2) Can you be slightly clearer in your methods by telling the reader that the way you arrived at 4840 data points is by taking each frame (121) in each of the 40 videos as a data point?

3) Could you clarify how exactly you made all the videos to be of the same length, this wasn't totally clear to me?

Small changes:

Page 3, line 85: I think you should refer to 'emotion categories' as opposed to mental representations to be clearer here for the reader.

Page 6, line 159: Was the ISI 'about' 300 ms or 'exactly' 300 ms? I.e., did you have a variable ISI or was this always 300ms?

Page 9, line 240: Remove the word 'mean' or 'refer to'.

6. PLOS authors have the option to publish the peer review history of their article (what does this mean?). If published, this will include your full peer review and any attached files.

Reviewer #1: **Yes: **Jeffrey Girard

Reviewer #2: No

---

## [Author Response · Author response to Decision Letter 0]

17 May 2022

Additional Editor Comments:

Dear Dr. Namba,

I have now received two reviews of the manuscript entitled The spatio-temporal features of perceived-as-genuine and deliberate expressions (PONE-D-21-33666) that you submitted to PLOS ONE. I was fortunate to secure reviews from two experts in the areas of facial expression analysis and emotional genuineness, both of whom provided strong insight to the strengths and weaknesses of the manuscript.

I read the original manuscript, and then again with comments provided by the reviewers. As you will see, the reviewers found the manuscript to be interesting, and the analytic techniques impressive. The topic is also relevant to the broad readership at PLOS One.

Both reviewers identified the academic merit of the manuscript, however noting that the clarity of the manuscript could be improved. Both reviewers provided guidance on improving the readability, which I encourage you to consider. There was disagreement as to the statistical & methodological rigor of the work, with Reviewer 1 taking issue with some analysis decisions. Both suggested major revisions. As such, I am rejecting the manuscript in its current form, and I invite you to resubmit in the form of major revisions.

You will find the full set of reviewer comments included below. To these I add my own two comments:

1) I would appreciate seeing a clear statement of theoretical contribution of this work. I read your 2021 Scientific Reports article with great interest. However, the 2021 work is only minimally reviewed, and readers may be left unclear as to what this work contributes to our understanding of genuine expressions. I recognize that there are differences with this work – OpenFace vs DeepLabCuts, Happy-Angry vs Surprise, ActionUnits vs landmarks, but these differences are methodological, rather than conceptual. I believe the core contribution relates to the perceptual nature of genuine expressions, but this should be more clearly contrasted from your previous works, along with those in the field.

Thank you for your comments. We have added a new paragraph with a more detailed description of the contribution of this work (Page 17, Lines 412–421). 

2) The stimuli presenting Genuine-vs-Deliberate expressions are central to the work, with both perceptual and motion analyses relying on their validity. Yet there is almost no discussion as to how these stimuli were produced, and whether the scenarios would be considered ‘genuine’. This presents a serious methodological oversight which would need to be robustly addressed in your revision.

Following to the editor’s recommendation, we have added story examples (Pages 5-6, Lines 150–153) and a more detailed explanation of and a description of the limitations to our selection of facial database (Page 5, Lines 138–144; Page 18, Lines 441–444).

We sincerely appreciate your reading and consideration of our paper. 

 

Comments to the Author

We are grateful for the reviewers’ excellent and extremely helpful comments. We have addressed all of the issues that the reviewers have highlighted, and we believe that the manuscript has been considerably improved as a result of these changes. Please let us know if further changes are required. We will be more than happy to alter the manuscript further.

Reviewer #1: 

PRAISE

This manuscript explored the interesting question of whether there are morphological and temporal differences between facial behaviors that are perceived to be genuine vs. deliberate. I liked that it examined perceptions of human facial behaviors rather than computer-generated facial behaviors and agree that this is an important validation step. I also liked the attempt to simultaneously consider morphology and temporal characteristics of the behavioral displays. I also thought the visualizations were nice and appreciated the data and materials being made open.

CONCERNS

1. The manuscript’s readability would be helped if the authors were more consistent in their use of terms. As just one example, multiple different terms are used to refer to the person producing a facial behavior (e.g., sender, encoder) or perceiving it (e.g., observer, decoder, perceiver).

Thank you for this comment. We have changed the terms to make them more consistent way (e.g., Page 1, Lines 48–49; Page 2, Lines 53-54…). We hope that these changes will help increase readability.

2. The methodological decision to look at onset plus and minus 2 sec creates some standardization but also erodes some of the ecological validity of the stimuli. If temporal characteristics matter for perception, as the authors contend, then why cut them partway through and therefore eliminate the ability of perceivers to see their duration, offset, and other temporal features?

We agree with Reviewer #1 in that the standardized onset and offset of facial expressions could potentially undermine ecological validity. Therefore, we must keep in mind that the conclusions drawn from our observations could be limited. Nevertheless, we needed to instruct the models to intentionally produce facial expressions within a certain time range to make it easier to compare perceived genuine and deliberate expressions at the expense of the natural and spontaneous time course of facial expressions. We believed that a certain degree of standardization (i.e., looking at onset plus and minus 2 sec) is desirable when presenting these facial datasets. However, as the reviewer mentioned, traditionally it is more desirable to consider offset as well (e.g., Schmidt et al., 2006: Movement Differences between Deliberate and Spontaneous Facial Expressions: Zygomaticus Major Action in Smiling), and recent studies have suggested that offset is important in determining authenticity (Horic-Aselin et al., 2020).

To clarify these points, we added the description of the potential limitations of our method in the Discussion section (Page 21, Lines 441–444) and provided more detailed explanation of the method of video recording in the Method section (Page 5, Lines 138–144).

3. Please provide a rationale for making the genuineness vs. deliberate ratings dichotomous. Why not have them rate this quality on a continuum? Forcing a choice between two alternatives throws away valuable information about degree (DeCoster et al., 2009) and introduces (or at least exacerbates) participants’ subjective thresholds as a source of variability.

Thank you for your comments. As you suggest, it would be more conservative to measure this authenticity as a continuous rating on two axes, genuine and deliberate (Dawel et al., 2017). Consequently, a better approach would be to provide a rating scale instead of a binary judgement. We have added this limitation to the Discussion section and cited your recommended work (Page 19, Lines 453-458).

4. The first paragraph in the Statistical Analysis section (page 6) was very difficult to understand. I think the authors calculated the percentage of participants that rated each stimulus as genuine and then used that quantity to select a subset of the stimuli most consistently rated as genuine vs. deliberate. But the descriptions and some words (“prescribed” and “judgment ratios”) were not adequately clear/explained.

Following the reviewer’s suggestion, we have modified our description for reader clarity (Pages 8–9, Lines 208–223). In addition to modifying this description, we have made major changes to the study actions; please refer to the response to the following comment (Comment #5).

5. Related to concern #4 above, I did not love the methodological decision to analyze 40 stimuli rather than all 180. I think this was done in order to compare behaviors that are perceived-by-most-as-genuine to perceived-by-most-as-deliberate rather than having more ambiguous behaviors. But this selection makes the results much less generalizable. We are not learning about most facial behaviors, we are learning about the 22% least ambiguous ones. If genuineness had been rated dimensionally (as in concern #3), the authors could have regressed it on various temporal features and included all 180 stimuli. If they used a mixed effects model, they also could have explicitly modeled encoder and decoder effects.

Thank you for this insightful comment. Following the reviewer’s suggestion, we used as many stimuli as possible for the analysis. We sincerely appreciate your comment and apologize for the discrepancy in the revised paper caused by conducting a detailed analysis from scratch (N = 180 → 148). We have drastically altered our statistical approach using the complete dataset. 

First, we used all of the data for anger and happiness that the current study used (N = 128) and confirmed that the same NMF spatial patterns were extracted as reported in the first draft. This can be interpreted as indicating the robustness of the NMF data obtained in this study.

Regarding the relationship between the decoders’ judgments of genuineness and identified NMF spatial patterns, we added a new mixed model which had explicitly modeled encoder and decoder effects. This logistic regression analysis provided us with results that are more easily interpretable for the spatio-temporal features of perceived-as-deliberate expressions (Page 8, Lines 198–207; Page 11, Lines 267–273; Page 13, Lines 309–313).

For the temporal features, we added “ambiguous” as a new condition for facial expressions that fell into the rating between as the intermediate between perceived-as-genuine and perceived-as-deliberate. By splitting δ into two, we succeeded in finding a generalization effect of increasing data and in separating the perceived-as-genuine effect from the perceived-as-deliberate effect (Pages 9-10, Lines 228–245).

We hope that our statistical approach has been improved drastically thanks to the reviewers’ detailed comments.

6. The authors acknowledge that OpenFace may be incorrect but do not provide any validation evidence such as comparing it to expert FACS coding on a subset of this specific dataset. Because we don’t know how much to trust the AU intensity estimates, we don’t know how much to trust the NMF components and these are really important variables in the paper.

Reviewer #1 is right. Although our previous study showed that OpenFace AU detection for Japanese also performs similarly to manual coding by a certificated FACS coder (Namba et al., 2021; r = 0.8), we have added further details to the method and limitation sections with the addition of this information (Page 7, Lines 184-186; Page 19, Lines 459-469). Furthermore, by adding new analyses (thanks to Comment #5), it is clear that the results of NMF are reliable, at least in the sense that they contribute to the decoder's judgment (Figs 2, 4). Thank you.

7. Please add some information to the Statistical Analysis section about you will be calculating 99% credible intervals at each value of t and how to interpret the figures (e.g., seeing whether delta=0 falls within that interval). Some of this information is currently in the figure captions only and it is very easy to miss.

Following the reviewer’s suggestion, we have added further explanations of credible intervals (Page 8, Lines 206–207; Page 10, Lines 244–245). 

8. In addition to plotting the 99% credible intervals at each value of t, I would be interested to see tests done on larger regions of t-space. For example, the authors could calculate the average delta score in the regions from t=0 to t=2000 in intervals of 250 or 500ms. They could also report probability of direction (pd) values for these regions in addition to credible intervals (Makowski et al., 2019).

Thank you very much for your thoughtful comment. We have calculated the average delta score at intervals of 500 ms. Following this, we extracted the 99% credible intervals and probability of direction (pd) values for these regions. For readability, we have provided this information in the Supplemental Tables. 

COMMENTS

1. The claim on page 1 line 50 that perceived-as-genuine expressions have more significant impacts on observer’s behavior than perceived-as-deliberate expressions is a leap that does not follow directly from the evidence presented earlier in the paper.

We have modified this paragraph to clarify the intended meaning (Pages 1–2, Lines 50–55).

2. Page 4 would be improved by providing an explicit rationale for the decision to look at anger and happiness rather than other possible emotions.

We have added a description identifying the rationale for looking at anger and happiness (Pages 3-4, Lines 102–104). 

3. On page 5, please briefly describe what the four emotional scenarios were for happiness and anger.

As the reviewer noted, we have added the example of stories (Pages 5-6, Lines 150–153).

4. Were there are quality or attention check questions to ensure that the participants were taking the task seriously? I’m not sure if this is a big issue in Japan, but in the US many crowdworkers do a poor job and this can be detected with some basic questions.

The experiment we employed this time took the form of contracting and hiring each person over time, so we expect the results to be similar in nature to those of a face-to-face experiments done online.

(There will be some degree of responsibility in that each participant will be linked to an answer and an ID, rather than just distributing links, as tends to happen in general online experiments.)

However, we acknowledged that response bias may be inevitable for online participants, and it is important to include such item checks. For this reason, we have provided a justification for Crowdworks Participants (Page 5, Lines 126–127) and added this information the discussion (Pages 18-19, Lines 449–453).

REFERENCES

DeCoster, J., Iselin, A.-M. R., & Gallucci, M. (2009). A conceptual and empirical examination of justifications for dichotomization. Psychological Methods, 14(4), 349–366. https://doi.org/10/bh86w7

Makowski, D., Ben-Shachar, M. S., Chen, S. H. A., & Lüdecke, D. (2019). Indices of effect existence and significance in the Bayesian framework. Frontiers in Psychology, 10. https://doi.org/10/ggfw2j

 

Reviewer #2: Thank you for inviting me to review this manuscript. The authors investigate spatio-temporal features of facial emotion expressions that help to distinguish perceived-as-genuine and perceived-as-deliberate (i.e., non-authentic) expressions. They do so by first presenting a series of dynamic facial emotion expression videos to 89 Japanese individuals and asking them to determine whether each expression is genuine or deliberate (completed for happy and angry expressions). They then reduced each expression video to its lowest dimensional features (as a function of the emotion and 'perceived as' conditions) using non-negative matrix factorisation. Both perceived-as-genuine and perceived-as-deliberate expressions showed protocol typical spatial action units, whilst deliberate expressions produced multiple facial movements in addition to these. Deliberate expressions also had faster onsets to peak expression and had a greater contribution of mouth opening movements.

The question being investigated here is novel, interesting and important. The techniques utilised to answer this question are in my opinion strong and in line with a number of recent studies applying dimensionality reduction techniques to facial emotion expressions. I commend the authors for this. I find the results very interesting and I also think they did a good job to put their findings in context as well as discuss the limitations (e.g., cultural differences that may exist and require further attention.

I have some comments on the write-up (mostly focused on the introduction) that I believe would strengthen the manuscript if rectified:

Introduction:

1) I feel the introduction could be tightened up a little bit so it really 'tells the story' for your reader to follow as to the rational for the current study. Some parts I struggled to follow the relevance of. For example, I wasn't sure I followed how the study is specifically related to the affective pragmatics theory and why this needs to be linked to an 'evolutionary account' that really had very little explanation. Perhaps you can build a story for the studies importance without these theories, especially as you don't refer back to this in the discussion, so I'm not sure exactly how relevant it is. Or alternatively, tighten up the explanation and refer back to it in the discussion.

Thank you very much for your thoughtful comments. We agree with this suggestion, and we have attempted to tighten up our introduction and omit these theories (Pages 1–2, Lines 47–57), referring back to this in the discussion section (Page 17, Lines 412–421). We hope that this change has improved the readability of the manuscript. Thanks again.

2) I think your design is great, whereby you are able to determine what spatio-temporal features are important in the human perception or labelling of genuine vs deliberate (as opposed to determining yourselves what a genuine or deliberate expression looks like) and I agree this is in line with the work by Rachel Jack and colleagues. However, I think you could make this point and emphasise the benefit of this kind of approach clearer for the reader. You could tighten up the explanation of Jack and co.'s method.

Thank you for your suggestion. In response to your comment, we have tightened up our presentation of Jack and her colleague’s method, and we have emphasized the specific benefit our approach brings relative to theirs in this section (Page 3, Lines 78–88)

3) My colleagues and I have investigated temporal features (mean speed) of happy, angry and sad expressions in posed and spontaneous contexts (Sowden et al., 2021; https://psycnet.apa.org/fulltext/2021-22419-001.html) which might also fit well into your introduction and/or discussion to support differences here. We found (in Experiment 1C) that speed of mouth opening and widening actions helped to differentiate between happy, angry and sad expressions only for posed and not spontaneous expressions, whilst speed of eyebrow widening movements was capable of differentiating these emotions for both posed and spontaneous expressions. This to me seems a nice finding alongside your findings, whilst your design has the nice element of looking at timecourses as opposed to our mean speed over entire expression period.

Thank you for mentioning these important reports in the literature. We have cited these works in the revised manuscript (Page 2, Lines 69–71; Page 16, Lines 382–388; Page 16, Lines 393–395). We are honored that the relationship with the proposed study has enriched our contribution.

Methods

1) I think the methods section would benefit from a little more explanation of what the NMF dimensionality reduction technique is and what it is essentially doing, for readers who do not know about this technique. On this topic, the authors might be interested in (and may wish to cite) an interesting paper by Chiovetti et al. (2018; https://jov.arvojournals.org/article.aspx?articleid=2678770) who seek to reduce various facial expressions down to their lowest dimensional organisation (regarding temporal properties of expressions) whereby emotion can be determined. They also compare NMF and a Fourier-based Anechoic Demixing Algorithm in doing so.

Thank you for noticing these important reports from the literature. We have added a simpler explanation for the NMF approach and added information on the Chiovetto’s work in the revised manuscript (Page 7, Lines 190–197).

2) Can you be slightly clearer in your methods by telling the reader that the way you arrived at 4840 data points is by taking each frame (121) in each of the 40 videos as a data point?

We have modified this paragraph to clarify our intention (Page 8, Lines 216–217).

3) Could you clarify how exactly you made all the videos to be of the same length, this wasn't totally clear to me?

In response to this comment, we added information about how we made all the videos the same length (Page 5, Lines 138–146).

Small changes:

Page 3, line 85: I think you should refer to 'emotion categories' as opposed to mental representations to be clearer here for the reader.

Accordingly, we have added a description in this paragraph (Page 3, Lines 79–81). 

Page 6, line 159: Was the ISI 'about' 300 ms or 'exactly' 300 ms? I.e., did you have a variable ISI or was this always 300ms?

Page 9, line 240: Remove the word 'mean' or 'refer to'.

We have added the term exactly and removed the word mean. Thank you for noticing these points (Page 6, Line 174, Page 12, Lines 294–297).

---

## [Decision Letter · Decision Letter 1]

23 Jun 2022

The spatio-temporal features of perceived-as-genuine and deliberate expressions

PONE-D-21-33666R1

Dear Dr. Namba,

We’re pleased to inform you that your manuscript has been judged scientifically suitable for publication and will be formally accepted for publication once it meets all outstanding technical requirements. Please see the minor suggested comments of R2, which you may wish to incorporate prior to proofing.

Kind regards,

Steven R. Livingstone

Academic Editor

PLOS ONE

Additional Editor Comments (optional):

Please see the minor suggested comments from R2 prior to your proofing.

Reviewers' comments:

Reviewer's Responses to Questions

**Comments to the Author**

1. If the authors have adequately addressed your comments raised in a previous round of review and you feel that this manuscript is now acceptable for publication, you may indicate that here to bypass the “Comments to the Author” section, enter your conflict of interest statement in the “Confidential to Editor” section, and submit your "Accept" recommendation.

Reviewer #1: All comments have been addressed

Reviewer #2: All comments have been addressed

2. Is the manuscript technically sound, and do the data support the conclusions?

Reviewer #1: (No Response)

Reviewer #2: Yes

3. Has the statistical analysis been performed appropriately and rigorously? 

Reviewer #1: (No Response)

Reviewer #2: Yes

4. Have the authors made all data underlying the findings in their manuscript fully available?

Reviewer #1: (No Response)

Reviewer #2: Yes

5. Is the manuscript presented in an intelligible fashion and written in standard English?

Reviewer #1: (No Response)

Reviewer #2: Yes

6. Review Comments to the Author

Reviewer #1: (No Response)

Reviewer #2: I am happy with responses to my comments and associated changes. Just three very minor comments remain:

1) When discussing the Sowden et al paper in the introduction, for clarity I would change the sentence to read: "Sowden et al demonstrated, using facial landmarks, that speed of facial movements differentiates deliberate expressions of anger, happiness and sadness."

2) Line 106: delete the words 'as to'.

3) Line 169-170: change sentence to read "If the participants responded with no questions, the main trials began."

7. PLOS authors have the option to publish the peer review history of their article (what does this mean?). If published, this will include your full peer review and any attached files.

Reviewer #1: **Yes: **Jeffrey Girard

Reviewer #2: No

---

## [Editor Report · Acceptance letter]

7 Jul 2022

PONE-D-21-33666R1 

The spatio-temporal features of perceived-as-genuine and deliberate expressions 

Dear Dr. Namba:

I'm pleased to inform you that your manuscript has been deemed suitable for publication in PLOS ONE. Congratulations! Your manuscript is now with our production department. 

Kind regards, 

on behalf of

Dr. Steven R. Livingstone 

Academic Editor

PLOS ONE